# Hot Spot TERT Promoter Mutations Are Rare in Sporadic Pancreatic Neuroendocrine Neoplasms and Associated with Telomere Length and Epigenetic Expression Patterns

**DOI:** 10.3390/cancers12061625

**Published:** 2020-06-19

**Authors:** Alexandra Posch, Sarah Hofer-Zeni, Eckhard Klieser, Florian Primavesi, Elisabeth Naderlinger, Anita Brandstetter, Martin Filipits, Romana Urbas, Stefan Swiercynski, Tarkan Jäger, Paul Winkelmann, Tobias Kiesslich, Lingeng Lu, Daniel Neureiter, Stefan Stättner, Klaus Holzmann

**Affiliations:** 1Department of Medicine I, Division: Institute of Cancer Research, Comprehensive Cancer Center, Medical University of Vienna, 1090 Vienna, Austria; a01305157@unet.univie.ac.at (A.P.); sarah_h.z@hotmail.com (S.H.-Z.); elisabeth.naderlinger@gmx.at (E.N.); anita.brandstetter@meduniwien.ac.at (A.B.); martin.filipits@meduniwien.ac.at (M.F.); 2Institute of Pathology, Paracelsus Medical University/Salzburger Landeskliniken (SALK), 5020 Salzburg, Austria; e.klieser@salk.at (E.K.); paul.winkelmann@stud.pmu.ac.at (P.W.); d.neureiter@salk.at (D.N.); 3Department of Visceral, Transplant and Thoracic Surgery, Medical University of Innsbruck, 6020 Innsbruck, Austria; florian.primavesi@i-med.ac.at (F.P.); s.staettner@icloud.com (S.S.); 4Regional Medical Directorate of the Province of Salzburg, Office of the Salzburg Provincial Government, Sebastian-Stief-Gasse 2, 5020 Salzburg, Austria; romana.urbas@salzburg.gv.at; 5Department of Surgery, Paracelsus Medical University, Salzburger Landeskliniken (SALK), 5020 Salzburg, Austria; stefan.swierczynski@gmail.com (S.S.); ta.jaeger@salk.at (T.J.); 6Department of Internal Medicine I & Institute of Physiology and Pathophysiology, Paracelsus Medical University/Salzburger Landeskliniken (SALK), 5020 Salzburg, Austria; t.kiesslich@salk.at; 7Yale Department of Chronic Disease Epidemiology, School of Public Health, School of Medicine, Yale Cancer Center, Yale University, New Haven, CT 06520-8034, USA; lingeng.lu@yale.edu; 8Department of Surgery, Salzkammergutkliniken, 4840 Vöcklabruck, Austria

**Keywords:** pancreatic neuroendocrine tumor, TERT promoter mutation, telomere length, tumor heterogeneity, pyrosequencing, microRNA, histone deacetylase

## Abstract

Cancer cells activate a telomere maintenance mechanism like telomerase in order to proliferate indefinitely. Telomerase can be reactivated by gain-of-function Telomerase Reverse Transcriptase (TERT) promoter mutations (TPMs) that occur in several cancer subtypes with high incidence and association with diagnosis, prognosis and epigenetics. However, such information about TPMs in sporadic pancreatic neuroendocrine neoplasms (pNENs) including tumor (pNET) and carcinoma (pNEC) is less well defined. We have studied two hot spot TPMs and telomere length (TL) in pNEN and compared the results with clinicopathological information and proliferation-associated miRNA/HDAC expression profiles. DNA was isolated from formalin-fixed paraffin-embedded (FFPE) tissue of 58 sporadic pNEN patients. T allele frequency of C250T and C228T TPM was analyzed by pyrosequencing, relative TL as telomeric content by qPCR. In total, five pNEN cases (9%) including four pNETs and one pNEC were identified with TPMs, four cases with exclusive C250T as predominant TPM and one case with both C250T and C228T. T allele frequencies of DNA isolated from adjacent high tumor cell content FFPE tissue varied considerably, which may indicate TPM tumor heterogeneity. Overall and disease-free survival was not associated with TPM versus wild-type pNEN cases. Binary category analyses indicated a marginally significant relationship between TPM status and longer telomeres (*p* = 0.086), and changes in expression of miR449a (*p* = 0.157), HDAC4 (*p* = 0.146) and HDAC9 (*p* = 0.149). Future studies with larger patient cohorts are needed to assess the true clinical value of these rare mutations in pNEN.

## 1. Introduction

Telomere maintenance mechanisms (TMMs) are one of the hallmarks through which cancer cells develop indefinite proliferation capacity. Hereby, increased telomerase activity (TA) represents the common type of TMM in most malignancies [1,2]. Telomerase is a ribonucleoprotein and reverse transcriptase, which adds units of the repeated telomere core sequence to the distal chromosomal ends [3]. Telomerase protein is encoded by the Telomerase Reverse Transcriptase (TERT) gene and its expression is generally the limiting factor for TA in somatic cells. TA becomes activated in cancer cells by mechanisms including chromosome rearrangements, epigenetics and TERT promoter mutations (TPMs) [4]. These TPMs are the most frequent non-coding mutations in cancer that can occur early during tumorigenesis [5]. Two such human cancer-associated hot spot TPMs are cytidine-to-thymidine changes at genomic loci Chr5:1,295,228 (C228T) and 1,295,250 (C250T), located upstream of TERT transcription and translation start [6,7]. 

Pancreatic neuroendocrine neoplasms (pNENs) are the second most common malignancies of the pancreas and mainly occur in elderly patients [8]. Their incidence of currently 1–2/100000 inhabitants in the United States and Europe is steadily increasing [9]. A recent multicenter study in Austria has shown 5 and 10 year survival rates after surgical resection of around 81% and 50%, respectively [9]. A large international cohort study validated the pNEN grading system to distinguish well differentiated pNET (G1, G2, G3) and poorly differentiated pNEC (only G3 by definition) [10]. Based on the secretion of peptide hormones, pNENs can be classified as functional and non-functional, with the majority of 60–85% being non-functional [11]. Moreover, pNENs can be associated with familial syndromes in approximately 10% of cases, while most pNENs are sporadic. TPMs in pNETs were recently found to be mainly associated to patients with hereditary syndromes, but not to sporadic cases [12]. In detail, 4 of 55 (7%) pNET patients showed exclusive single TPMs with C228T, but no C250T alteration and 3 of 4 (75%) TPM cases occurred in patients with hereditary syndromes, such as multiple endocrine type 1 (MEN1) and Von Hippel-Lindau (VHL). 

Here we analyzed the TPM status at hot spots C250T and C228T of a patient cohort diagnosed with sporadic pNEN, which was recently studied for expression of epigenetic factors, such as proliferation-associated miRNAs and expression of histone deacetylases (HDACs) [13]. TPMs were detected in pNENs with varying T allele frequencies and were found associated with parameters for tumor progression, with higher telomere content and changes in miRNA/HDAC expression. 

## 2. Results

### 2.1. TPM Analysis and Clinicopathological Characteristics of Patients 

Clinical and histopathological parameters of 58 sporadic pNEN patients included in this study are summarized and grouped according to wild-type and C250T and C228T TPMs (Table 1 and Table 2). Fifty-two pNET were 90% of pNEN cases studied. Parameters included gender, age, relapse status and survival of patients, and tumor characteristics such as size, localization, TNM and T-ENETS staging, grading, lymphatic and vasculature invasion, proliferation and hormone activity. No significant correlations with *p* < 0.05 between TPM status and clinicopathological parameters were observed (Table 1). However, some of the parameters showed an association with the TPM classified pNEN patients, e.g., advanced age and metastasis (*p* = 0.24–0.28). These observed trends may indicate a relation between TPMs and pNEN tumor progression. 

C250T and C228T TPMs were detected by pyrosequencing of PCR amplified DNA from the TERT promoter region (Figure 1). PCR product analyses of DNA samples isolated from formalin-fixed paraffin-embedded (FFPE) tumor tissue slices and cell line controls resulted amplicon size similar to expected 128 bp (Figure 1A). Pyrosequencing of PCR amplicons allowed quantification of the cytidine-to-thymidine change at hot spot C250T and C228T genomic loci (Figure 1B). Control cell lines with known C250T and C228T TPM status were analyzed for C and T allele frequencies together with tumor tissue samples of pNEN patients (Figure 2). TPMs were identified with T allele frequencies >1% for 5 of 58 (9%) pNEN patients, four cases at C250T, and one case at both C250T and C228T. TPMs were identified in both pNEN subcategories, 4 of 52 (8%) pNET and 1 of 6 (17%) pNEC. T allele frequencies of DNA isolated from adjacent FFPE tumor tissue slices varied in 4 of 5 TPM cases more as compared to control cell line replicates (Figure 2). In detail, T allele frequencies of adjacent tissue slices and of tumor cell lines resulted in coefficient of variation (CV) values with mean of 101% and 10%, respectively. This extreme variability may indicate tumor cell and/or TPM heterogeneity in pNEN. Five pNEN cases with TPMs identified are listed with detailed clinicopathological parameters including the evaluation of tumor cell content by histological validation (Table 2). Tumor cell content of the FFPE tumor tissue from 5 TPM cases varied between 65 and 93%. These high tumor cell contents support the notion of common TPM heterogeneity within pNEN cells. Notably, 4 of the 5 pNEN patients were female with C250T TPM. The tumor with TPM of a male patient was the only one found with C228T TPM in addition to C250T TPM. In this studied cohort of patients diagnosed with sporadic pNEN, C250T was found more frequently than C228T TPM. According to histological validation case #12 showed the lowest tumor cell content and the highest variability regarding T allele frequencies among the 5 TPM cases. In contrast, cases #5 and #33 showed the highest tumor cell content and the lowest variability regarding T allele frequencies, with low mutated T allele frequencies up to 5%. 

### 2.2. Association of TPM Status with Telomeric Content, miRNA and HDAC Expression

Telomeric content (TC) is related with telomere length (TL) and was analyzed by qPCR of DNA samples isolated from FFPE tumor tissue relative to a cell line control with long telomeres (Figure 3). 

Expression levels of proliferation associated miRNAs and immunohistochemical expression patterns of members of the four HDAC classes were published recently for a large subset of the pNEN cases [13]. No significant correlations of TPM with TC and the expression of epigenetic factors were assessed (Table 3). However, a trend was observed for increased TC indicating longer TL in pNEN with TPM as compared to wild-type (*p* = 0.086). Moreover, mean TPM T allele frequency of pNEN tissue was associated with TC (Figure 3). In detail, high allele content of only C250T TPM was found to be associated with an increased TC in pNEN. This relationship supports the notion of TPM and TL heterogeneity in sporadic pNEN cancer cells with increased TC in case of higher numbers of TPM tumor cells. 

TPM were found marginally associated with changes in miRNA and HDAC expression profiles. TPM cases demonstrated a trend for increase in miR449a (*p* = 0.157) and decrease in miR132-3p (*p* = 0.264), HDAC4 (*p* = 0.146) and HDAC9 (*p* = 0.149) expression (Table 3). HDAC4 and HDAC9 are less expressed/absent in the nucleus of pNEN cells with TPM as compared to wild-type (Figure 4). Staining for HDAC9 is, in general, weak compared with the other HDACs and moreover missing in the nucleus of healthy tissue from TPM positive pNEN patients but not TPM wt pNEN patients. 

## 3. Discussion

This retrospective study examined the TPM status and TL of FFPE tissue from 58 patients diagnosed with sporadic pNEN and compared the results with clinicopathological parameters and available miRNA and HDAC expression profiles [13].

TPM status was analyzed by pyrosequencing of pNEN tissue with high tumor cell content and 5 of 58 patients (9%) were identified with the C250T and one of them additionally with the C228T TPM allele. Each TPM creates a de novo binding site for transcription factors, which are recruited to the mutant but not the wild-type promoter to activate TERT transcription and TA as TMM [14]. To our knowledge, this is the first time that TPM T allele frequencies have been assessed in pNEN using pyrosequencing, an approach applied by several others in various tumors including glioma, melanoma, laryngeal, gallbladder and gastric cancer [15,16,17,18]. However, all these studies deployed cut-off limits for TPM detection above that one used for this pNEN study. 

Initial studies on TPM in various tumor types did not report any hot-spot TPM in a cohort of 68 pNET cases using Sanger sequencing and described pNET and other tumor types with such low incidences that they almost always originate from tissues with relatively low rates of self-renewal [2,19]. Another study with a similar sample size which also used Sanger sequencing led to the assumption that TPMs in pNET are rare with 7% (4 of 55) cases with the C228T, but not the C250T allele, and mainly occur in 3 of 4 patients with hereditary syndromes [12]. This study confirmed only 1 of 55 (2%) sporadic pNET patient cases with C228T TPM, in contrast to our finding of 5 of 58 (9%) pNEN cases with C250T TPM identified by pyrosequencing. Generally, C228T mutation is more prevalent than C250T among various malignancies and their presence is mutually exclusive [20]. In conclusion, the current data support the notion that TPM in sporadic pNEN patients and in pNEN patients with hereditary syndromes are both similar rare events but with specific prevalence of C250T and C228T TPM, respectively. A possible reason for the identified increase of the TPM incidence rate by this study might be that pyrosequencing is known to be more sensitive than Sanger sequencing and other methods [21]. T allele increase above the low cut-off limit of 1% by pyrosequencing has already been rated as a mutation after another positive repetition of the experiment. 

A recent pan-cancer analyses of whole genomes reported cancer and their matching normal tissues across 38 tumor types, including a cohort of 85 patients (30 female, 55 male, median age 59) with neuroendocrine carcinomas from the pancreas [22]. This, and an accompanying study, demonstrated that TPM in pNEN were absent [23]. Instead, 23% of pNEN cases showed ATRX/DAXX truncation mutations previously associated with alternative lengthening of telomeres (ALT) as TMM [24,25]. Whether any non-defined TMMs may be operating in the majority of pNEN cases and other tumor types needs further studies [2]. 

The trend for long telomeres identified in 9% TPM pNEN cases supports the notion that some of the other pNEN cases with short telomeres may not use ALT or an always active TMM in all tumor cells. An accompanying study of the pan-cancer analyses of whole genomes demonstrated intra-tumor heterogeneity is generally widespread and tumor subclones contain drivers, such as the TPM activated TERT gene, that are under positive selection [26]. As already mentioned, in the pNEN cohort of the pan-cancer analyses with 85 patients no TPM was identified. However, for tumors including pNEN the timing analyses suggest that driver mutations often precede diagnosis by many years and highlight chances for early cancer detection. Further studies using more sensitive methods such as deep sequencing will still need to elucidate to what extent rare TPMs may play a role in pNEN cancer genome analyzes. 

T allele frequencies from 4 of 5 TPM pNEN cases varied e.g., with a range between 0 and 76% when pyrosequencing analyses were replicated with adjacent FFPE tumor tissue and high tumor cell content of >60%. This finding supports the notion of TPM heterogeneity in pNEN cases by existence of tumor subclones with and without TPM at the time of diagnosis. 

Known risk factors for poor survival after resection of pNETs include increased tumor size, presence of metastasis, vascular and lymphogenic invasion, positive lymph nodes of the primary tumor, T-Stage and non-functioning tumors [9,27]. The five identified TPM cases were not found significantly related to any of the clinicopathological and epigenetic parameters in this small sample of pNEN patients. The statistical analysis was likely underpowered to find significant differences between the two groups TPM and wild-type, since the TPM group was small (n = 5). However, there was a non-significant association of TPM cases with metastasis and advanced patient age. Interestingly, no such trend was observed for overall and disease-free survival of the patients. Several studies of other tumor types identified that TPMs predict worse patient outcome, e.g., [17,28,29,30]. TPMs are not only prognostic factors, but also predictors of radiotherapy resistance in gliomas [31] and biomarkers for clinical non-invasive testing of bladder cancer diagnosis and surveillance [32]. Targeting of non-coding TPMs remains an unresolved challenge, but recent findings raise the possibility that telomerase inhibition currently investigated in clinical trials might be an effective intervention in cells with TPM [5,33]. This clinical potential of TPMs in cancer needs to be studied and exploited for pNENs. 

Tumors with TPM were found on average approximately 1 cm smaller than those without TPM, which would be contradictory with the proposed more aggressive behavior of TPM cases. ALT positive primary pNENs are known to be associated with larger tumor size, higher T stage, aggressive clinical behavior and poor survival [34,35,36]. Therefore, it may be speculated that TPMs are more frequently found in smaller tumors, while ALT is more commonly found in larger tumors. Furthermore, TPM cases were associated with expression of specific epigenetic factors. TPM cases demonstrated trends for changes in specific miRNAs and HDACs, with factors known to be associated with TMMs [4]. HDAC9 positively regulates the ALT pathway and was found increased in ALT cells [37]. The five TPM cases showed a trend (*p* = 0.149) for decreased/absent HDAC9 expression in the nucleus of pNEN cells. This finding supports the notion that absence of HDAC9 expression may exclude ALT activity in case of TPM activated TA in pNEN. However, because of low case numbers studied and no standardized schemes for the evaluation of immunohistochemical staining, there is currently no strong evidence if HDAC expression profiles might be a reliable surrogate marker for TPM status. Future studies with larger number of pNEN patients are needed to examine the relation of TPM with clinical parameters, histological features and patient survival. 

## 4. Materials and Methods

### 4.1. Clinical and Pathological Characterization of Patients

The studied cohort included 58 pNEN patients with archived formalin-fixed paraffin-embedded (FFPE) tissue specimens as described recently [13] with minor modifications. Three pNEN cases that were surgically resected in 2015 replaced 2 cases due to limitations in availability of tissue. In brief, all cases were characterized for clinicopathological parameters including gender, age, tumor size, TNM staging according to the AJCC/UICC 2017 classification [38], ENETS 2006 classification, invasion of lymphatic vessels (L), vascular invasion (V), grading according to WHO 2017 definition [39] with the Ki67 associated proliferation index, resection status (R), overall survival (OS), disease-free survival (DFS), hormone activity and hereditary syndrome association as described [13]. Survival was defined as time in months from surgery to last follow-up/death (OS) or last follow-up/recurrence (DFS). This retrospective study with anonymized samples was conducted following the national and institutional guidelines of the Paracelsus Medical University Salzburg as well as in accordance with the declaration of Helsinki (1964). All analyses on human pNEN samples were approved by the local ethics committee (Ethikkommission für das Bundesland Salzburg, Postfach 527, A-5010 Salzburg; 415-EP/73/408-2014) 19 May 2014. Hereditary syndromes (MEN1, VHL, etc.) were ruled out by pre-screening based on clinical information followed by genetic counselling and genetic sequencing in cases with suspicious clinical findings or positive familial history. For all 5 TPM cases there was no positive familial history for genetic tumor syndromes and no clinical suspicion of MEN1, VHL, etc., therefore, no genetic testing was necessary/performed. 

### 4.2. Cell Line Controls

Tumor cell lines were obtained from the American Type Culture Collection (ATCC, Manassas, VA, USA). Growth conditions for T98G and U2OS cell lines were described recently [40]. U87MG cells were grown at 37 °C under 5% CO2 in Dulbecco’s Modified Eagle Medium (DMEM) with 10% fetal bovine serum (FBS), 10 µL/mL non-essential amino acids (NEAA) and 2 µL/mL pyruvate.

### 4.3. Isolation of DNA 

DNA of FFPE tumor tissue and cell lines was extracted using the nexttec 1-Step DNA Isolation Kit for Tissue and Cells (Biozym Scientific GmbH, Germany) according to the manufacturer’s manual. FFPE tissue was deparaffinized with xylene [41]. Based on the tumor area (cm^2^) up to four 10 μm FFPE sections were used for DNA isolation, resulting in eluated DNA in a volume of 100 µL. Tumor content was determined taking into account the area of the tumor tissue (in percentage of the whole corresponding slide) and its cellularity (in percentage of the tumor tissue). Cases number 5, 12, 15, 33, and 58 were identified with TPM and DNA isolation was carried out three times using adjacent FFPE slices. DNA concentration was measured using Qubit fluorometer (Thermo Fisher Scientific, Vienna, Austria) and Qubit dsDNA High Sensitivity Assay Kit (Thermo Fisher Scientific, Vienna, Austria) according to the manufacturer’s instructions with detection range 20 pg to 200 ng. Thus, 2 μL isolated DNA samples were used for each measurement. DNA samples were stored at −80 °C. 

### 4.4. TERT Promoter Mutation (TPM) Analysis by Pyrosequencing

DNA was amplified with specific primers for TERT promoter region using the PyroMark PCR Kit (Qiagen, Hilden, Germany) according to the manufacturer’s instructions. In detail, the forward primer sequence was adapted from a publication [42]: 5′-CGTCCTGCCCCTTCACCT-3′ and the reverse primer sequence was biotinylated: 5′-[Biotin]-GGGGCCGCGGAAAGGAA-3′. Primers were ordered with the PyroMark Custom Assay (Qiagen, Hilden, Germany). Isolated DNA was diluted to concentration of 2 ng/µL and 5 µL of this dilution were used per PCR, resulting in 10 ng DNA per PCR. For samples with DNA concentrations below 2 ng/µL, 5 µL of the isolated DNA were directly used per PCR, resulting in DNA input amounts lower than 10 ng ranging between 2.4 and 5.1 ng for 4 pNEN cases analyzed. A non-target control (NTC) containing nuclease-free water instead of DNA was included for each PCR experiment. Cycling was carried out on a Peqstar thermocycler (VWR, Germany) with the following conditions: 95 °C for 15 min; 45 cycles with 94 °C for 30 s, 66 °C for 30 s and 72 °C for 30 s; 72 °C for 10 min and 4 °C for hold. Size of PCR products was analyzed on 2% agarose gel by gelelectrophoresis prior to pyrosequencing. Also, 50 bp DNA Ladder (Thermo Fisher Scientific, Vienna, Austria) was used as a marker.

Pyrosequencing of the TERT promoter hot spot mutations C250T and C228T was carried out on the PyroMark Q24 MDx device (Qiagen, Hilden, Germany) using PyroMark Gold Q24 reagents (Qiagen, Hilden, Germany) according to the manufacturer’s instructions with following minor modifications. 10 μL PCR product was used per sequencing reaction with sequencing primer 5′-ACCCCGCCCCGTCCCGA-3′ (Qiagen, Hilden, Germany). Instead of heating up samples to 80 °C on a heating block, samples were placed in an incubator set to 82 °C. Analysis was carried out with PyroMark Q24 2.0.8 software using AQ function, the dispensation order GACGTCATGTCAGTCAGTCTAC and the sequence to analyze CCCCTYCCGGGTCCCCGGCCCAGCCCCYTCCG. Y marks variable position, being either cytosine (C) or thymidine (T). The manufacturer of the pyrosequencer describes the sensitivity with 2% mutation and 98% wild-type (wt). FFPE tumor tissue sample was considered mutated if T allele frequency at either hot spot position was ≥2% or considered wt if the T allele frequency was ≤1%. PCR and pyrosequencing of the respective tissue samples were repeated to confirm mutation. 

The glioblastoma cell lines T98G and U87MG were used as positive controls for C250T and C228T mutation, respectively [43] and DNA from these control cell lines was included in each PCR and pyrosequencing experiments. 

### 4.5. Immunohistochemistry and Processing for Markers of HDACs (1-6, 8-11 and Sirt1)

TPM case number 58 was analyzed by immunohistochemistry (IHC) for markers of HDACs 1–6, 8–11 and Sirt1 as described previously for tissue microarrays (TMA) [13]. Fifty-seven cases have been recently analyzed on five TMA to ensure comparability of the IHC signals with all applied primary antibodies described [44]. In brief, absolute quantitative scoring data of each case and HDAC staining was carried out semiquantitatively. The number of stained cells was counted, and the intensity of the staining was assessed. According to the datasheets of HDAC antibodies, cytoplasmic (only HDAC4, HDAC5, HDAC8, and HDAC10) and/or nuclear (all HDACs) expression pattern was separately evaluated. Finally, an immunoreactivity score (range: 0–300) was calculated by multiplying the scores for intensity (0–3) and the stained cells (extensity, 0–100%). 

### 4.6. Relative Telomere Length Analysis by qPCR

Relative telomere length (TL) was determined as telomeric content (TC) by qPCR as described [45] with minor modifications. In detail, telomere (T) repeat and single (S) copy gene qPCR assays in 25 µL volumes with 500 nM primers each were performed in triplicates on an ABI PRISM 7500 Fast Sequence Detection System with 7500 v2.3 software (Applied Biosystems, Foster City, CA, USA) and GoTaq qPCR master mix (Promega, Walldorf, Germany). Ribosomal protein lateral stalk subunit P0 (RPLP0/36B4) was chosen as S gene. Standard curves for T and S qPCR were generated by using serial two-fold dilutions (8 ng/μL to 0.0625 ng/μL) of genomic DNA from U2OS cell line as a long telomere control with telomere length >20 kbp. U2OS DNA concentrations were calculated for each DNA sample using standard curves with average cycle threshold (Ct) results for T and S qPCR. Normalized TC was calculated as T/S ratio. Upper and lower error bars of normalized TC values were calculated by a worst case estimation of adding the coefficient of variations for U2OS DNA concentrations of T and S qPCR results. 

### 4.7. Statistical Analysis

Either Chi-square or Fisher’s exact test was appropriately used for categorical variables. Non-parametric Wilcoxon rank sum test was performed for continuous variables. Kaplan-Meier method with log-rank test was used to test for difference in survival. R packages (version 3.6.1 [46]) were used for statistical analyses. The median of individual miRNA expression levels was used as the cut-off for classifying patients into two miRNA expression groups (low or high). A p value less than 0.05 at the two-sided level was considered statistically significant.

## 5. Conclusions

This study identified hot-spot TPMs with the highest incidence known of 9% in sporadic pNEN using pyrosequencing as a high sensitivity method that allows quantitation of mutated T alleles. TPMs were found with extreme varying T allele frequencies in adjacent tumor tissue specimens. This variability may indicate TPM heterogeneity within pNEN. Identified associations of TPMs in pNEN with clinicopathological parameters such as tumor progression, telomere length and epigenetic factors provide important information for future studies with larger patient samples. 

## Figures and Tables

**Figure 1 cancers-12-01625-f001:**
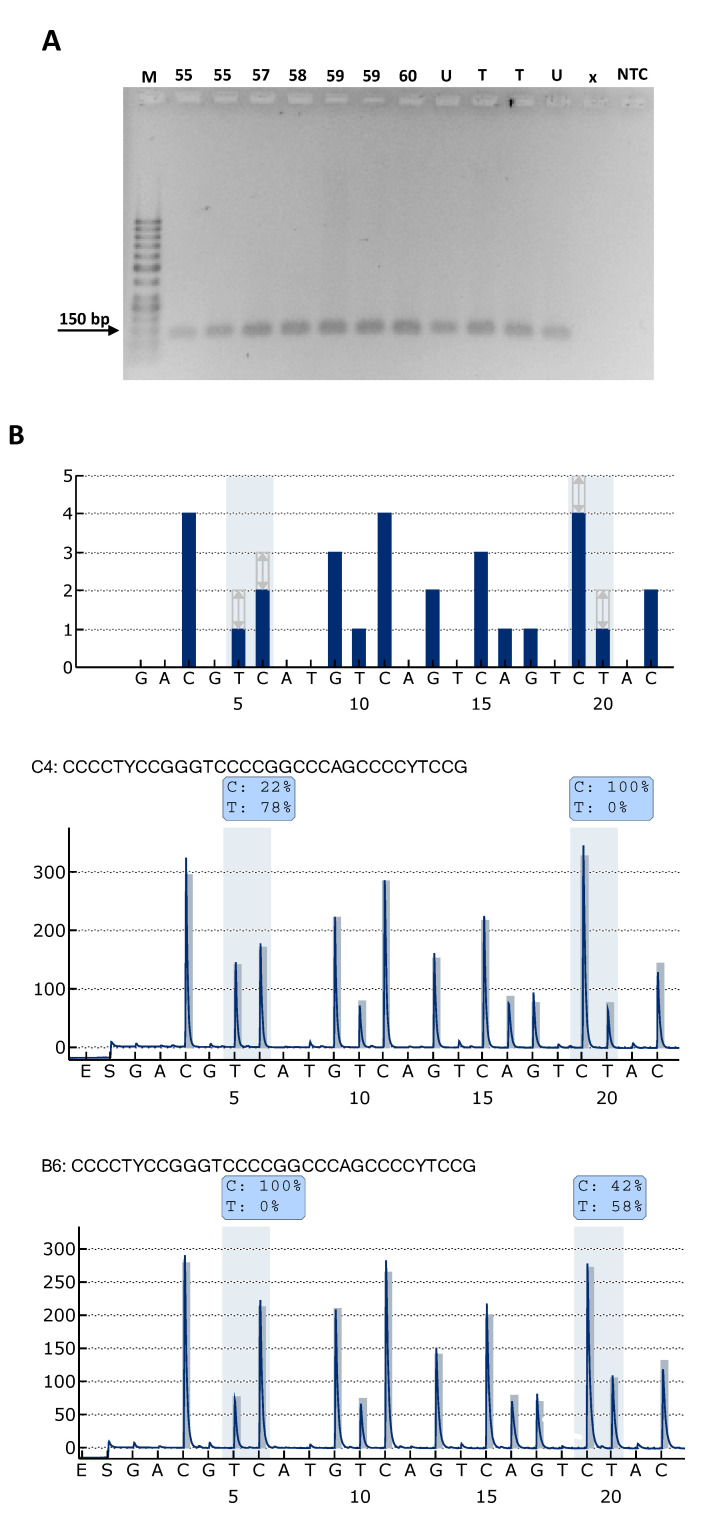
Analyses of T and C allele frequencies for C250T and C228T TPMs by PCR and pyrosequencing. (**A**) Representative result of PCR products analyzed by 2% agarose gel electrophoresis. DNA was isolated from FFPE tumor tissue and cell line controls (U: U87MG, T: T98G) with known T allele frequencies. Numbers indicate the patient IDs from FFPE tumor tissue samples analyzed. M: 50bp DNA marker, NTC: non-target control for PCR, X: empty. (**B**) C250T and C228T TPM detection by pyrosequencing. (top panel) Histogram shows at y-axis the number of nucleotides incorporated and at the horizontal axis the dispensation order with theoretical representation of the expected peak patterns. The grey background with arrows indicates the quantified C and T bases of the two hot spot TPMs C250T and C228T at dispensation order positions 5/6 and 19/20, respectively. Representative pyrogram results of TPM control cell lines T98G (middle panel) and U87MG (lower panel). Nucleotides were added successively in the pyrosequencing reactions following the sequence to analyze as displayed above the pyrogram. Each adequate nucleotide included generates a light signal shown as a peak. Background signals were obtained by addition of enzyme (E) and substrate (S) of pyrosequencing reaction, and the first control nucleotides G and A that are not present in the sequence to analyze. The peak heights in the pyrogram marked by gray background reflect the ratio of cytosine (C) to thymine (T) at each hot spot TPM and are used for calculation of C and T allele frequencies shown in blue boxes for C250T (left) and C228T (right).

**Figure 2 cancers-12-01625-f002:**
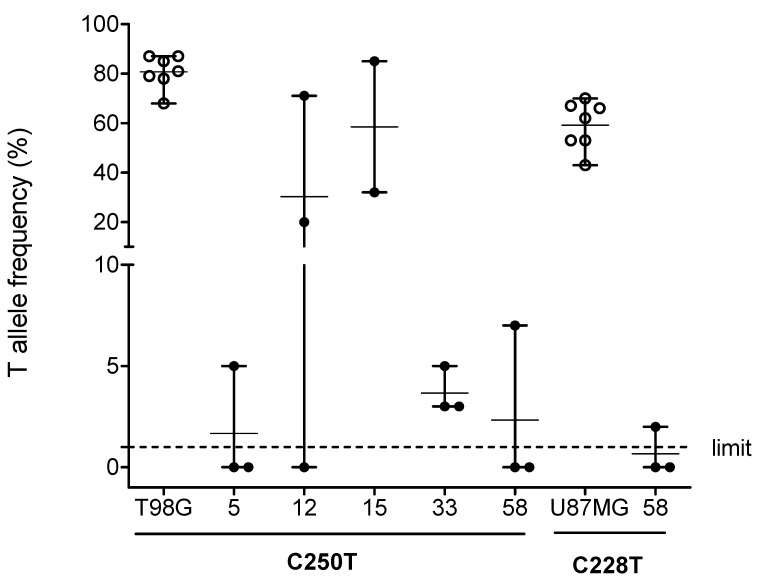
T allele frequencies of C250T and C228T TPMs from pNEN tissue specimens and control cell lines. Numbers indicate IDs of 5 pNEN patients identified with T allele frequencies >1% as limit for detection. T98G and U87MG control cell lines with known TPM status. Filled dots represent, for each patient, the results of pyrosequencing replicates of DNA isolated from adjacent FFPE tumor tissue slices. Open dots represent replicates of control cell lines. Error bars represent mean with range.

**Figure 3 cancers-12-01625-f003:**
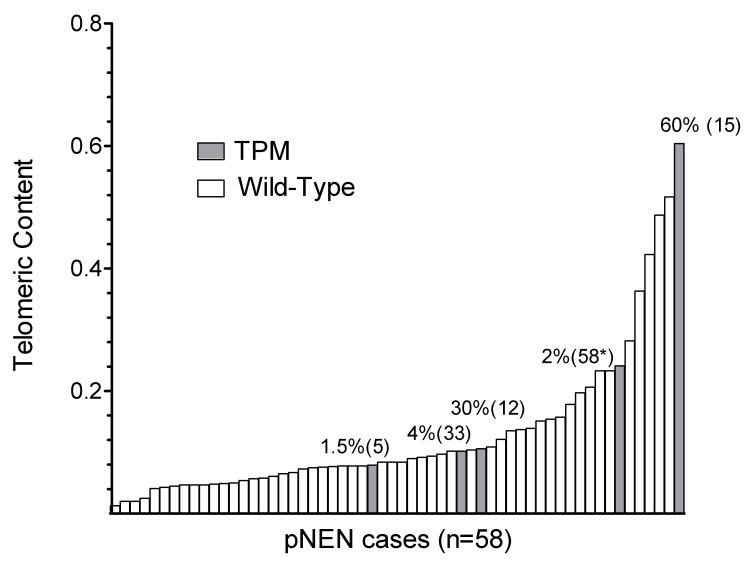
Telomere length and TPM status. Telomeric content was determined as T/S ratio by qPCR relative to control cell line U2OS with long telomeres defined as 1. Relative telomere lengths of pNEN cases are shown as waterfall blot. TPM cases are marked by filled bars and shown with T allele frequencies and case numbers in brackets.

**Figure 4 cancers-12-01625-f004:**
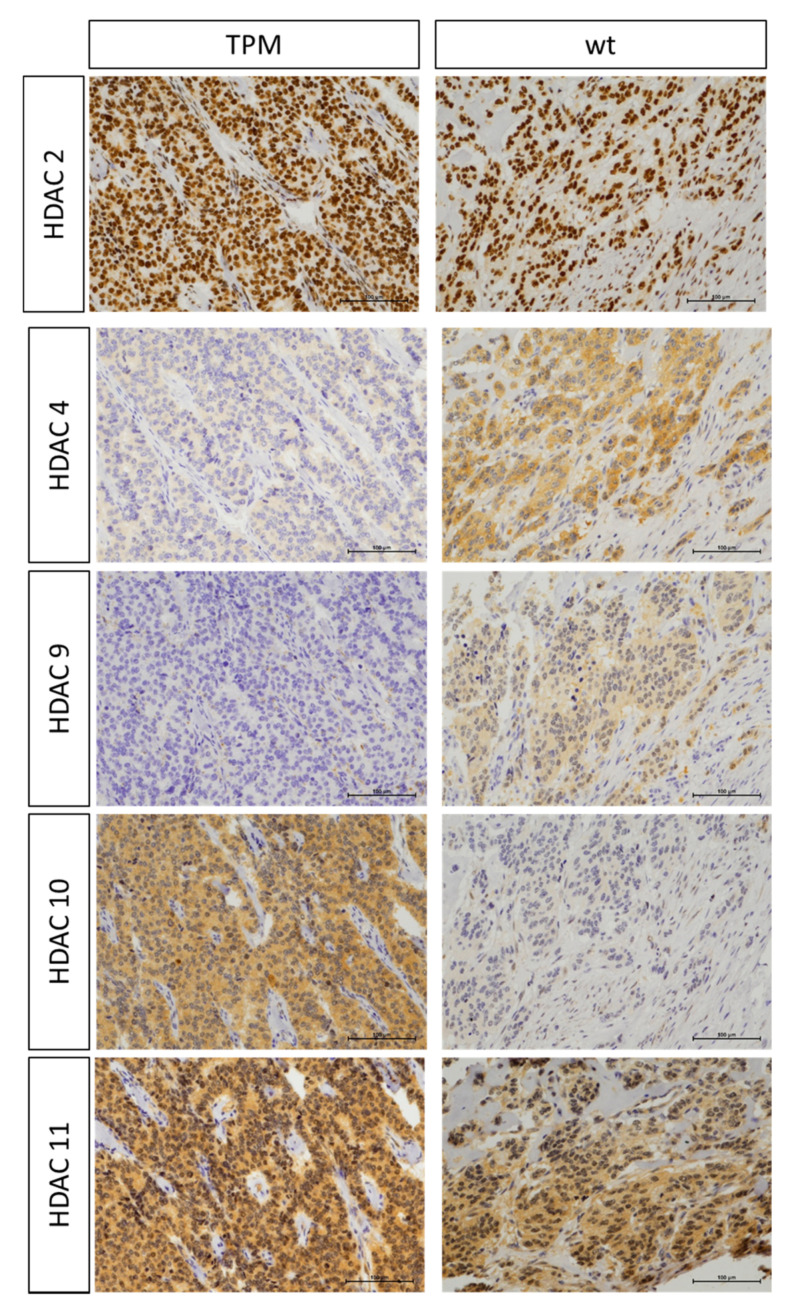
Immunohistochemistry (IHC) analysis of HDAC expression. Representative IHC staining of TPM case 12 and wild-type (wt) case 8 with antibodies for indicated HDACs are shown. Scale bar: 100 µm. With regard to HDAC4 and HDAC9, a trend towards lower/absent expression in the TPM cases was observed (both HADC class IIa). In contrast, a similar expression of HDACs was found in the other HDAC classes. There were some exceptions, such as in this comparison in the expression of HDAC10, but these exceptions did not cause significant differences or trends. Original magnification 200×.

**Table 1 cancers-12-01625-t001:** Clinicopathological features of sporadic pNEN patients grouped according to TPM status. *

Parameters	Cases			*p* Value
	with Data, *n* (%)	Wild-Type *n* = 53 (91%)	C250T/C228T *n* = 5 (9%)	
Gender	58 (100)			0.536
Male	25 (43)	24 (45)	1 (20)	
Female	33 (57)	29 (55)	4 (80)	
Age, years; mean ± SD	58 (100)	61.2 ± 14.4	68.4 ± 11.4	0.283
Size, cm; mean ± SD	57 (98)	2.85 ± 2.38	1.85 ± 1.25	0.361
Localization	58 (100)			0.613
Caput	23 (40)	22 (42)	1 (20)	
Corpus	10 (17)	9 (17)	1 (20)	
Cauda	22 (38)	19 (36)	3 (60)	
Unknown	3 (5)	3 (6)	0	
TNM-Staging (AJCC/UICC 2017)	58 (100)			
T-Category 1/2/3/4		19/13/16/5	3/0/1/1	0.433
N0/1		37/16	3/2	0.641
M0/1		44/9	3/2	0.237
M1a/1b/1c		5/3/1	2/0/0	1.00
T-ENETS (2006)	58 (100)			
1/2/3/4		19/13/16/5	3/0/1/1	0.434
Invasion of lymphatic vessels (L)	58 (100)			1.00
L1	7 (12)	7 (13)	0 (0)	
L0	51(88)	46 (87)	5 (100)	
Vascular invasion (V)	58 (100)			0.433
V1	6 (10)	5 (9)	1 (20)	
V0	52 (90)	48 (91)	4 (80)	
Grading (WHO 2017)	58 (100)			0.694
NET G1	30 (52)	27 (51)	3 (60)	
NET G2	20 (34)	19 (36)	1 (20)	
NET G3	2 (4)	2 (4)	0 (0)	
NEC G3	6 (10)	5 (9)	1 (20)	
R status	52 (90)			1.00
R0	47 (90)	43 (90)	4 (100)	
R1	5 (10)	5 (10)	0 (0)	
Overall survival (OS)	51 (88)	47 (89)	4 (80)	
OS months, median (range)		82 (6–232)	77 (9–110)	0.890
Disease-free survival (DFS)	50 (86)	46 (87)	4 (80)	
DFS months, median (range)		77 (0–194)	56 (9–110)	0.950
Ki67-index	58 (100)	53	5	0.657
Mean ± SD		10.7 ± 20.8	16.6 ± 20.9	
Hormonal activity	44 (76)			0.327
Yes	4 (9)	3 (8)	1 (25)	
No	40 (91)	37(93)	3 (75)	

* Data represents numbers and (percentage per group) unless otherwise stated. M status 1a = only liver, 1b = extrahepatic, 1c = liver and extrahepatic.

**Table 2 cancers-12-01625-t002:** Detailed clinicopathological data of pNEN patients with TPMs.

Parameters	Case Number				
	5	12 *	15	33	58
TPMs	C250T	C250T	C250T	C250T	C250T and C228T
T allele frequency range (%)	0–5	0–71	31–85	1–5	0–11 (C250T) 0–3 (C228T)
Tissue tumor cell content range (%)	90–95	60–70	75–80	90–95	85–90
Gender (male/female)	f	f	f	f	m
Age (years)	55.7	84.7	59.7	62.8	79.3
Tumor size (cm)	4.00	2.50	0.95	1.20	0.60
Localization	cauda	caput	cauda	corpus	cauda
TNM–Staging (AJCC/UICC 2017)					
T-category	4	3	1	1	1
N-status	1	1	0	0	0
M-status (a/b/c)	1a	1a	0	0	0
T-category (ENETS 2006)	4	3	1	1	1
Invasion of lymphatic vessels (L)	no	no	no	no	no
Vascular invasion (V)	no	yes	no	no	no
Grading (WHO 2017)	NEC G3	NET G2	NET G1	NET G1	NET G1
R status	0	n.a.	0	0	n.a.
Ki67 (%)	54.6	5.7	2.0	1.7	1.2
Hormone syndrome association	no	n.a.	yes	no	no

* pNET was incidental diagnosis in the context of an autopsy, n.a., not available.

**Table 3 cancers-12-01625-t003:** Correlation of TPM status with telomeric content, histone deacetylase (HDAC) expression and miRNA expression.

Parameters *	Cases												*p* Value
	TPM Wild-Type						TPM Mutant C250T/C228T						
	*n*	mean	median	SD	min	max	*n*	mean	median	SD	min	max	
HDAC1 ncl	43	153	150	75	5	300	5	159	128	93	40	285	0.852
HDAC2 ncl	44	193	170	66	90	300	5	181	190	50	113	238	0.778
HDAC3 ncl	43	102	88	84	0	285	4	73	35	96	10	213	0.434
HDAC4 ncl	41	32	30	26	0	83	5	17	10	24	0	60	0.146
HDAC4 cyt	41	72	70	48	0	213	5	47	15	52	10	128	0.297
HDAC5 ncl	42	166	170	62	3	300	5	175	190	48	98	213	0.862
HDAC5 cyt	42	96	98	44	10	190	5	117	113	59	50	190	0.521
HDAC6 cyt	44	105	113	48	20	213	5	112	120	81	10	190	0.728
HDAC8 ncl	46	10	5	19	0	113	5	4	0	6	0	10	0.347
HDAC8 cyt	43	15	10	23	0	130	3	17	0	29	0	50	0.665
HDAC9 ncl	46	4	5	6	0	30	5	1	0	2	0	5	0.149
HDAC10 ncl	40	100	120	71	0	250	5	64	60	62	10	160	0.384
HDAC10 cyt	40	138	165	74	5	300	5	119	135	70	10	190	0.549
HDAC11 ncl	42	132	128	56	3	238	5	124	113	42	75	190	0.704
miR132-3p	51	0.2038	0.0633	0.4358	0.0041	2.7602	4	0.0709	0.0290	0.0816	0.0138	0.2118	0.264
miR145-5p	51	1.0603	0.5649	1.1733	0.0453	4.7951	4	0.6500	0.5434	0.5141	0.1491	1.3640	0.427
miR183-5p	51	0.0741	0.0186	0.1900	0.0003	1.3351	4	0.0185	0.0188	0.0060	0.0098	0.0266	0.961
miR34a-5p	51	0.2545	0.1618	0.2536	0.0483	1.4541	4	0.1627	0.1386	0.0900	0.0681	0.3065	0.466
miR449a	51	0.0020	0.0008	0.0062	0.0001	0.0452	4	0.0040	0.0023	0.0040	0.0007	0.0106	0.157
Telomeric content (TC)	53	0.121	0.084	0.112	0.013	0.517	5	0.226	0.106	0.221	0.079	0.604	0.086

* ncl nucleus. cyt cytoplasm.

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
