# Peer review of "Hot Spot TERT Promoter Mutations Are Rare in Sporadic Pancreatic Neuroendocrine Neoplasms and Associated with Telomere Length and Epigenetic Expression Patterns"

_cancers, 2020, doi:10.3390/cancers12061625_

Round 1
Reviewer 1 Report
The authors well described the TPM status and TL of sporadic pNET and compared the results with clinicopathological parameters.
I have only some advices:
In introduction I think you should also consider in addition to your refence n.9(Eur J Surg Oncol 2019, 45, 198-206) also the data of this importat recent series:
Competitive Testing of the WHO 2010 versus the WHO 2017 Grading of Pancreatic Neuroendocrine Neoplasms: Data from a Large International Cohort Study.
Rindi G, Klersy C, Albarello L, Baudin E, Bianchi A, Buchler MW, Caplin M, Couvelard A, Cros J, de Herder WW, Delle Fave G, Doglioni C, Federspiel B, Fischer L, Fusai G, Gavazzi F, Hansen CP, Inzani F, Jann H, Komminoth P, Knigge UP, Landoni L, La Rosa S, Lawlor RT, Luong TV, Marinoni I, Panzuto F, Pape UF, Partelli S, Perren A, Rinzivillo M, Rubini C, Ruszniewski P, Scarpa A, Schmitt A, Schinzari G, Scoazec JY, Sessa F, Solcia E, Spaggiari P, Toumpanakis C, Vanoli A, Wiedenmann B, Zamboni G, Zandee WT, Zerbi A, Falconi M. Neuroendocrinology. 2018;107(4):375-386.
As method update grading and staging systems to classify studied neoplasms. In particular use WHO2019 for grading (and not WHO 2010) distinguishing well differentiated tumors (G1, G2, G3) and poorly differentiated carcinomas (only G3 by definition); refer also to indications of 8th edition of AJCC for staging.
In all text and tables moreover use the word pNET only to define pancreatic well differentiated tumours; use pNEN (neuroendocrine neoplasm) to define general category of pancreatic neuroendocrine neoplasms including both well and poorly differentiated neoplasms; use pNEC (neuroendocrine carcinoma) only for poorly differentiated neoplasms. In particular among your G3 studied cases you should specify if NET or NEC.
The results concerning the molecular analysis are well presented, however, data regarding immunohistochemistry for markers of HDACs (1-6, 8-11 and Sirt1) are a bit confusing.
I suggest authors to:
- Clearly indicate in the methods section the cut-off and all other parameters utilized for IHC
- Is IHC a reliable surrogate for TPM status? Please discuss more in deep this topic
- If possible, add an additional figure of IHC analysis.
Molecular analysis: FFPE tumor tissue sample was considered mutated if T allele 292 frequency at either hot spot position was > 1%. Is this cut off value a gold standard? If so please add references otherwise explain how authors applied this cut-off.
Author Response
Response to Reviewer 1 Comments:
The authors well described the TPM status and TL of sporadic pNET and compared the results with clinicopathological parameters.
I have only some advices:
Response: Many thanks to the reviewer for the suggestions to improve the manuscript. Line numbers refer to the version with track changes.
Point 1. In introduction I think you should also consider in addition to your refence n.9(Eur J Surg Oncol 2019, 45, 198-206) also the data of this importat recent series:
Competitive Testing of the WHO 2010 versus the WHO 2017 Grading of Pancreatic Neuroendocrine Neoplasms: Data from a Large International Cohort Study.
Rindi G, Klersy C, Albarello L, Baudin E, Bianchi A, Buchler MW, Caplin M, Couvelard A, Cros J, de Herder WW, Delle Fave G, Doglioni C, Federspiel B, Fischer L, Fusai G, Gavazzi F, Hansen CP, Inzani F, Jann H, Komminoth P, Knigge UP, Landoni L, La Rosa S, Lawlor RT, Luong TV, Marinoni I, Panzuto F, Pape UF, Partelli S, Perren A, Rinzivillo M, Rubini C, Ruszniewski P, Scarpa A, Schmitt A, Schinzari G, Scoazec JY, Sessa F, Solcia E, Spaggiari P, Toumpanakis C, Vanoli A, Wiedenmann B, Zamboni G, Zandee WT, Zerbi A, Falconi M. Neuroendocrinology. 2018;107(4):375-386.
As method update grading and staging systems to classify studied neoplasms. In particular use WHO2019 for grading (and not WHO 2010) distinguishing well differentiated tumors (G1, G2, G3) and poorly differentiated carcinomas (only G3 by definition); refer also to indications of 8th edition of AJCC for staging.
In all text and tables moreover use the word pNET only to define pancreatic well differentiated tumours; use pNEN (neuroendocrine neoplasm) to define general category of pancreatic neuroendocrine neoplasms including both well and poorly differentiated neoplasms; use pNEC (neuroendocrine carcinoma) only for poorly differentiated neoplasms. In particular among your G3 studied cases you should specify if NET or NEC.
Response 1: We now updated our collective according to WHO 2017 guidelines for grading (NET G1/G2/G3 or NEC) and according of 8th edition of AJCC for staging. We included the new data in Table 1 and Table 2. In addition, information of Rindi et al. 2018 was included in introduction section (lines 67-69). Reference Amin et al. 2017 of 8th edition of AJCC for staging was included in methods section (line 274). Reference for grading was included (line 276). All text in the manuscript was corrected for specific use of pNEN, pNET and pNEC as recommended.
Point 2. The results concerning the molecular analysis are well presented, however, data regarding immunohistochemistry for markers of HDACs (1-6, 8-11 and Sirt1) are a bit confusing.
I suggest authors to:
- Clearly indicate in the methods section the cut-off and all other parameters utilized for IHC
- Is IHC a reliable surrogate for TPM status? Please discuss more in deep this topic
- If possible, add an additional figure of IHC analysis.
Response 2: IHC details were described in the methods (lines 342-349). Discussion of HDAC IHC as possible surrogate marker for TPM status was included (Lines 262-265). Additional figure of IHC analysis was included as Figure 4 and referred in the text (line 174).
Point 3. Molecular analysis: FFPE tumor tissue sample was considered mutated if T allele 292 frequency at either hot spot position was > 1%. Is this cut off value a gold standard? If so please add references otherwise explain how authors applied this cut-off.
Response 3: The manufacturer of the pyrosequencer describes the sensitivity with 2% mutation and 98% wt (https://www.qiagen.com/us/service-and-support/learning-hub/technologies-and-research-topics/pyrosequencing-resource-center/technology-overview/). We applied in this study for detection of mutations a cut-off >1% or >=2%, and we technically replicated all positive results to confirm mutations and to exclude false-positives as described in methods (lines 330-334). This low cut off value was not chosen by others as described by references in discussion (lines 198 to 199). We add some more information to methods to explain how we applied the low cut-off (lines 330 to 331).

Reviewer 2 Report
This review discusses the discovery of TERT promoter mutations in a subset of sporadic pancreatic neuroendocrine tumors. I have some questions and suggestions.
* Do TPMs have clinical relevance? Are they targetable with existing or "in-the-pipeline" therapy? This should be addressed in the Discussion.
* The authors may wish to mention that their statistical analysis was likely underpowered to find significant differences between the two groups, since the TPM group was small (n=5).
* Case 5 (with TPM) has a Ki67 of 54.6%. Did the authors confirm this is a neuroendocrine tumor and not a neuroendocrine carcinoma?
* The caption for Figure 1 spends a couple sentences explaning how pyrosequencing works. Is this necessary?
* Table 3 adds very little to the manuscript and could perhaps be supplemental.
* Lines 173-176. The authors mention theirs appears to be the first study to use pyrosequencing to detect TPMs. Why is that? They mention the benefit to using this technique, but are there any drawbacks?
* Lines 191-192. The words "In detail" are probably not necessary, since this sentence is in fact not very detailed.
* Line 195. Why mention neuroendocrine carcinomas here? They are not the focus of this projet; neuroendocrine tumors are.
* The authors note that patients "with suspicious clinical findings or positive family history" underwent additional screening to rule out syndromes. Did this apply to any of the 5 patients with TPMs?
* Line 299-300. This sentence is unclear. Please rephrase.
Author Response
Response to Reviewer 2 Comments:
This review discusses the discovery of TERT promoter mutations in a subset of sporadic pancreatic neuroendocrine tumors. I have some questions and suggestions.
Response: Many thanks to the reviewer for the suggestions to improve the manuscript. Line numbers refer to the version with track changes.
Point 1: * Do TPMs have clinical relevance? Are they targetable with existing or "in-the-pipeline" therapy? This should be addressed in the Discussion.
Response 1: The clinical relevance of TPMs and their potential as therapeutic targets was included in discussion (lines 245-251).
Point 2:* The authors may wish to mention that their statistical analysis was likely underpowered to find significant differences between the two groups, since the TPM group was small (n=5).
Response 2: Statement added to discussion (lines 241-242).
Point 3:* Case 5 (with TPM) has a Ki67 of 54.6%. Did the authors confirm this is a neuroendocrine tumor and not a neuroendocrine carcinoma?
Response 3: As recommended by another reviewer, we now updated our collective according to WHO 2017 guidelines for grading (NET G1/G2/G3 or NEC) and according of 8th edition of AJCC for staging. We included the new data in Table 1 and Table 2. As suspected, case 5 and other cases are therefore NEC. All text in the manuscript was corrected for specific use of pNEN, pNET and pNEC as recommended.
Point 4:* The caption for Figure 1 spends a couple sentences explaning how pyrosequencing works. Is this necessary?
Response 4: We think the explanation of pyrosequencing and pyrogram is important to possibly better understand the higher sensitivity as compared to other standard methods. The caption about detailed pyrosequencing description was reduced (lines 137 to 139).
Point 5:* Table 3 adds very little to the manuscript and could perhaps be supplemental.
Response 5: We agree, but we want to keep this information in the main article as it may be interesting for readers to get detailed parameters of the identified rare TPM cases, including the new grading and staging categories.
Point 6:* Lines 173-176. The authors mention theirs appears to be the first study to use pyrosequencing to detect TPMs. Why is that? They mention the benefit to using this technique, but are there any drawbacks?
Response 6: We possibly describe the first study to use pyrosequencing to detect TPMs in pNEN (line 196-198). The manufacturer of the pyrosequencer describes the sensitivity with 2% mutation and 98% wt (https://www.qiagen.com/us/service-and-support/learning-hub/technologies-and-research-topics/pyrosequencing-resource-center/technology-overview/). We applied in this study for detection of mutations a cut-off >1% or >=2%, and we technically replicated all positive results to confirm mutations and to exclude false-positives as described in methods (lines 330-335). Drawbacks of this technique might be more laborious and time-consuming handling with specific equipment and expensive costs compared to other more routine methods like Sanger sequencing.
Point 7:* Lines 191-192. The words "In detail" are probably not necessary, since this sentence is in fact not very detailed.
Response 7: Sentence corrected accordingly (line 213).
Point 8:* Line 195. Why mention neuroendocrine carcinomas here? They are not the focus of this projet; neuroendocrine tumors are.
Response 8: As answered in point 2, we have now classified the collective according to the current WHO guidelines. This paper covers all pancreatic neuroendocrine neoplasias including neuroendocrine carcinomas: NET G1/G2/G3 and NEC. Discussion of NEC was not changed (Line 218).
Point 9:* The authors note that patients "with suspicious clinical findings or positive family history" underwent additional screening to rule out syndromes. Did this apply to any of the 5 patients with TPMs?
Response 9: In all of these 5 TPM mutant cases there was no positive familial history for genetic tumor syndromes and no clinical suspicion of MEN1, VHL, etc., therefore, no genetic testing was necessary / performed. We added this information to the methods (lines 286-288).
Point 10:* Line 299-300. This sentence is unclear. Please rephrase.
Response 10: The sentence was rephrased (lines 340-342).
